# Porous Tantalum Acetabular Cups in Primary and Revision Total Hip Arthroplasty: What Has Been the Experience So Far?—A Systematic Literature Review

**DOI:** 10.3390/biomedicines12050959

**Published:** 2024-04-25

**Authors:** Evangelia Argyropoulou, Evangelos Sakellariou, Athanasios Galanis, Panagiotis Karampinas, Meletis Rozis, Konstantinos Koutas, George Tsalimas, Elias Vasiliadis, John Vlamis, Spiros Pneumaticos

**Affiliations:** 1Department of Orthopaedics and Traumatology, University General Hospital of Patras, 26504 Patras, Greece; k.koutas.cy@gmail.com; 2Department of Orthopaedic Surgery, KAT General Hospital, National & Kapodistrian University of Athens, 14561 Athens, Greece; vagossak@hotmail.com (E.S.); athanasiosgalanis@yahoo.com (A.G.); karapana@yahoo.com (P.K.); mrozhs@gmail.com (M.R.); georgetsalimas@yahoo.com (G.T.); eliasvasiliadis@yahoo.gr (E.V.); jvlamis@email.com (J.V.); spirospneumaticos@gmail.com (S.P.)

**Keywords:** tantalum, bone loss, primary total hip arthroplasty, revision total hip arthroplasty, acetabular component, hip, acetabulum, acetabular cup

## Abstract

Background: The global population, especially in the Western world, is constantly aging and the need for total hip arthroplasties has rocketed, hence there has been a notable increase in revision total hip arthroplasty cases. As time has passed, a considerable developments in science and medicine have been attained which have also resulted in the evolution of both surgical techniques and implants. Continuous improvements have allowed large bore bearings to be utilized which provide an increased range of motion, with ameliorated stability and a very low rate of wear. The trend for almost the last two decades has been the employment of porous tantalum acetabular cups. Several studies exist comparing them with other conventional methods for total hip arthroplasties, exhibiting promising short and midterm results. Methods: The Preferred Reporting Items for Systematic Reviews and a Meta-Analysis (PRISMA) were used to identify published studies in a comprehensive search up to February 2023, and these studies were reviewed by the authors of the article. Specific rigorous pre-determined inclusion and exclusion criteria were implemented. Results: Fifty-one studies met our inclusion criteria and were involved in the systematic review. Sixteen studies examined postoperative clinical and radiological outcomes of using a tantalum cup in primary and revision total hip arthroplasty, whilst four biomechanical studies proved the superiority of tantalum acetabular components. Five articles provided a thorough comparison between tantalum and titanium acetabular cups, while the other studies analyzed long-terms results and complication rates. Conclusions: Porous tantalum acetabular cups appear to be a valuable option in revision total hip arthroplasty, providing clinical improvement, radiological stability, and promising long-term outcomes. However, ongoing research, longer follow-up periods, and careful consideration of patient factors are essential to further validate and refine the use of tantalum in various clinical scenarios.

## 1. Introduction

As the population is steadily growing older, hip joint-related problems have also proliferated and the majority of them need to be addressed surgically. The most severe and most common of these is hip arthritis, a disease that was not treatable surgically until the middle of the 20th century. A significant revolution in the 20th century was the development of the total hip arthroplasty (THA), which was heralded as the “operation of the century”, featuring a beneficial impact on the patients’ quality of life and their daily needs [1,2]. The prevalence of this surgical repair in the US population was 2.34% in 2010. Due to the increase in life expectancy and constant socioeconomic progress, the popularity of this specific surgery has increased dramatically. One of the most significant complications though, is the mechanical failure of the procedure, which requires a revision surgery (rTHA), a salvage procedure that involves several risks, such as infection that can progress to sepsis, stress on the body, longer exposure to anaesthesia, bleeding, and increased costs owing to extended duration of hospitalization and treatment [3,4]. Over 1,000,000 arthroplasties are performed worldwide at a growing rate, with predictions doubling this rate doubling over the next two decades. Following the rise in THA, there has also been a 43% to 70% increase in the frequency of rTHA in recent years [1].

As time has gone by, there has been an enormous development in science and medicine, leading to an evolution in the surgical techniques and materials used for bone implants. The first type of implants that were used were ceramic components, with their major negative aspects being their sensitivity to correct placement and their susceptibility to fracture. The trend for the last decade has been the employment of porous tantalum acetabular cups. Various studies have been carried out, comparing them with other conventional methods and implants in both primary and revision total hip arthroplasties [5].

Tantalum acetabular cups were initially introduced in 1997. Firstly, they appeared as an alternative to conventional titanium acetabular shells due to their longer lifespan [6]. While conventional porous materials, such as titanium, seemed to stabilize the acetabulum, tantalum implants promised acetabular stability through bone formation. The irregular surface of the component improves the integration into the host bone, by enhancing the structural and functional connection with tissue infiltration and adhesion through a process called osseointegration. As a result, tantalum implants are regarded as having some unique structural advantages over the competition [7,8].

In terms of the revision total hip arthroplasties, the utilization of porous tantalum implants has become a promising alternative. Different studies have indicated good midterm results without radiolucency, lower polyethylene wear, and a reduction in acetabular migration in contrast to other implants [6,8,9,10].

Tantalum acetabular cups have exhibited promising outcomes so far, but further investigation is essential to compare their effectiveness over the titanium ones. This paper aims to investigate the potency of tantalum acetabular components in primary and revision THAs, augmenting orthopaedic surgeons’ knowledge in their course of action when dealing with challenging cases. To the best of our knowledge, this paper is the first systematic review that comprises all studies concerning the employment of porous tantalum in primary and revision total hip arthroplasties.

## 2. Materials and Methods

The Preferred Reporting Items for Systematic Reviews and Meta-Analysis (PRISMA) guidelines were utilized in this systematic review, while no ethical approval was needed, and the review was registered in Insplay (INPLASY202430042). The studies that were identified and included in the study were written exclusively in English and formulated to contrast survivorship of patients porous tantalum acetabular implants with titanium acetabular implants in individuals having primary or revision total hip arthroplasty surgery. A succinct and systematic literature search was carried out for papers published in the following electronic literature databases: MEDLINE/Pubmed, Google Scholar, Web of Science and Embase. The research was carried out by analyzing papers published up to October 2023. Keyword search terms were: tantalum AND titanium AND hip arthroplasty. Reference lists from articles that met the inclusion criteria were further investigated (Table 1).

Pertinent papers were picked out for inclusion based on the following predetermined eligibility criteria:Reporting on human patients undergoing primary or revision total hip arthroplasty;Direct comparison between tantalum acetabular cups and conventional titanium acetabular cups employed in total hip arthroplasty;Radiological evaluation (cup migration, osteointegration);Clinical (functional scores, need for subsequent revision, patient-reported outcomes);Postoperative complications.

The search utilizing the aforementioned keywords yielded 450 (442 + 8) articles in total, until the 15 October 2023. The papers were scrutinized for duplication with resulting number of articles to 213, whilst 158 studies remained after non-English and irrelevant-to-our-topic papers were excluded. The authors separately evaluated the titles and abstract of each outcome, and those that were plainly extraneous and/or failed to be pertinent to the pre-decided inclusion criteria (*n* = 83) were removed. The remaining seventy-five (*n* = 75) papers were further investigated for patently apposite studies that indubitably met the inclusion criteria, deducting a further 24 trials. The full-texts of the remaining fifty-one (*n* = 51) articles were individually reviewed by the authors of this paper, who concurred that all of the studies were without-bias and pertinent to this summary. A flow chart of the systematic literature search according to PRISMA guidelines is presented bellow (Figure 1). 

## 3. Results

### 3.1. Clinical and Radiological Results

Sixteen studies examined the postoperative clinical and radiological results from studies that included the use of tantalum cups in both primary and revision THAs. Concerning the clinical results, the most commonly utilized scores were the HHS, OHS, and WOMAC scores, and all of the studies showed a significant improvement in all of them. Additionally, when exclusively taking into account monoblock tantalum cups [12,15,34], those scores upsurged. When compared to other porous materials or titanium, porous tantalum featured better clinical results [21,27] and with the employment of antiprotrusion cage, those results were even better [50]. Regarding radiological outcomes, no evidence of radiolucencies was observed in five of the studies, whilst acetabular cup placement was almost anatomical [60]. A study by Ayers et al. [27] revealed no noteworthy variations in terms of proximal migration between the porous tantalum and titanium acetabular implants, while various papers suggested that when dealing with severe acetabular bone defects, porous tantalum demonstrated excellent radiological outcomes [11,13,18,30,55]. Also, a small percentage of aseptic loosening, acetabular protrusion, and minor acetabular gaps filled by the last follow-ups were reported in some cases.

### 3.2. Mechanical Stability—Osseointegration and Biomechanical Studies

In terms of mechanical stability, four of the biomechanical studies proved the superiority of porous tantalum acetabular components. After contrasting tantalum with other porous metal components in vitro [19], and to titanium in vivo, tantalum was found to provide a greater primary stability at higher loads than titanium [51]. In addition, component firmness and polyethylene wear were meliorated by adding porous tantalum beads to the joint’s liner [29]. Moreover, in rTHA with severe acetabular defects, tantalum appeared to offer acceptable early migration with radiostereometric analysis and also an improvement in acetabular fixation with inferior screws [40]. Furthermore, concerning revision of the acetabular conmponents, three studies reported excellent stability [24]. More specifically, with the utilization of cup–cage constructs, the results were stable with a good potential for bone ingrowth [61] and the initial stability proved to be sufficient [26]. Lastly, in comparison with multiple types of implants, the porous tantalum trabecular metal implants exhibited higher osseointegration outcomes [41].

### 3.3. Safety and Complications

Fourteen articles contemplated the complication rates of porous tantalum use in rTHAs and only three assessed this in primary arthroplasties. In primary THAs, no incidence of infection or dislocation existed [38], and only 2% reported minor complications [16], except for one case that was revised owing to a deep infection after a monoblock THA [31]. On the other hand, the most salient complications in revision surgeries were infection and dislocation, with the rate of failure being higher in major bone loss defects cases [46]. Regarding infection, cases with hematogenous infection were scarce [30] and only 2–4% of the patients needed revision [44] with cup removal. Furthermore, a lower incidence of failure and subsequent infection was detected when tantalum was employed in patients with a periprosthetic joint infection [25]. The most common complication seemed to be dislocation [47], featuring a percentage ranging from 10% to 15% [55], indicating malposition and a small femoral head size (28 mm) as principal risk factors [48]. Furthermore, a study by Brüggemann et al. [49] showed that, in patients with stable tantalum acetabular cups, the amount of tantalum in their blood was low. The investigated articles propounded the utilization of jumbo acetabular cups with tantalum for improved outcomes, as they exhibited lower complication rates in general [22], as well as the use of larger femoral heads [39] or dual mobility constructs [42].

### 3.4. Long-Term Results

Eight out of the fifty-one articles examined and analyzed outcomes for longer follow-up periods, with the results favouring the tantalum implants. The reports so far indicated excellent pain relief and good functional outcomes and patient satisfaction over ten years [34]. Tantalum can be considered as a viable alternative option in pelvic discontinuity [44,55] and in patients with failed cage reconstruction with bone allografts [58]. In general, the studies reported positive long-term outcomes [31], with good implant survivorship [40]. Only one review paper by Rambani et al. [54], comparing tantalum with titanium acetabular shells, found a small advantage in short- to medium-term follow-up but the long-term results were promising for tantalum components. Nonetheless, further longer and multicentre studies are necessary [39].

### 3.5. Tantalum vs. Titanium Acetabular Cup

A rigorous comparison of complications, survivorship, and clinical outcomes between tantalum and titanium acetabular cups was developed in five of the articles (Figure 2). Depending on the type of acetabular defect, no statistically significant discrepancy was observed in implant survival and proximal migration when tantalum was utilized in primary or revision THA [27,35], but it appeared to be more valuable in revision surgeries in which there was a moderate-to-severe acetabular deficiency [17] as it may provide a greater primary stability at higher loads [51]. Broadly, porous tantalum implants are characterized by a lower risk of failure and contamination, providing promising long-term results, but little advantage in short- to medium-term follow-up [54].

### 3.6. Overall Comparison

#### 3.6.1. Positive Outcomes

Across various studies, porous tantalum acetabular cups demonstrated positive results in terms of clinical improvement, radiological stability, and survivorship.

#### 3.6.2. Application in Revision Cases

Tantalum is firmly regarded as a valuable option in revision total hip arthroplasty, specifically in cases with moderate-to-severe acetabular deficiency.

#### 3.6.3. Long-Term Success

Concerning tantalum acetabular cups, studies examining results with longer follow-up periods consistently reported excellent long-term outcomes, with improvements in patient-reported outcomes, clinical scores, and radiographic stability.

#### 3.6.4. Complications

Whilst complications, such as dislocation, infection, and aseptic loosening do exist, they are generally reported at low rates, and many studies highlight the importance of careful patient selection.

#### 3.6.5. Biomechanical Stability

The biomechanical studies reported that tantalum may offer meliorated stability and osseointegration, supporting the positive clinical outcomes observed.

## 4. Discussion

Total hip arthroplasties have gained momentous popularity over the last few decades, while the increase in the aging population has contributed materially to this growth. The increased number of THAs performed and the more aged patients undergoing surgery has led to considerable rise in the number of postoperative complications [1]. Continuous technological advancements have triggered the development of a variety of implants, with porous tantalum acetabular cups being favoured for approximately the last two decades [4,5]. This systematic review elaborates upon both the clinical and radiological outcomes of tantalum acetabular implants in both primary THA and revision THA.

Porous tantalum acetabular cups, according to the literature, present a higher coefficient of friction against cancellous bone, an element that augments their solidity and survival figures [30]. Simultaneous porous monoblock tantalum acetabular implants have been demonstrated to ameliorate the survivorship of primary cementless THAs. As for titanium cups, they are associated with aseptic loosening and high revision rates in the long-term [4,6]. The advantages of tantalum cups are inherent in the fact that they present a three-dimensional porous surface, with an average pore size of 550 μm and a porosity of 75–80% [9]. Concomitantly, even more pivotal is the fact that the modulus of resilience is 3Gpa, which is betwixt the cortical and subchondral bone [8]. Tantalum acetabular shells offer an advantageous design with an irregular surface that improves the osseointegration procedure, providing augmented stability even in major bone defects [6]. Unger et al. [11] suggested in 2005 that tantalum cups feature excellent radiological bone apposition and bone graft incorporation and might be suitable for the revision total hip arthroplasty (rTHA), but further investigation was requisite. In 2009, Li et al. [16] illustrated that tantalum implants offer a strong capacity for bone conduction and bone inducement. Several biomechanical studies from the last few years support the clinical results, like Menenghini et al. [19], who proved that tantalum implants exhibit better mechanical stability in vitro compared to other metal components. Furthermore, Solomon et al. [40] supported the employment of tantalum cups in revision of the acetabular component when severe bone loss is apparent and proposed inferior screws for even better fixation. Lastly, Nebergal et al. [29] conducted a radiostereometric analysis and corroborated that with the addition of tantalum bead in the cup liner, stability and polyethylene wear is substantially improved.

Implants’ survivorship is a key factor, and particularly in revision THA cases, where major bone defects are present and any additional revision surgery can be exceedingly arduous. Cassar-Gheiti et al. [52] in a cohort study of 59 patients reported impeccable tantalum implant survivorship and favourable clinical results in mid-term outcomes. In the same year, Concina et al. [55], in a clinical study with long-term results, indicated a Kaplan–Meier survivorship rate of 100%. Moreover, the large size of tantalum acetabular components increases the survivorship of the implant, when contrasted with traditional cage allografts. Jenkins et al. combined the utilization of tantalum cups with augments in rTHA cases with severe bone defects and received 97% implant survivorship, while also maintaining a satisfactory hip function [36].

Acetabular bone defect cases are exacting to embark on and are ordinarily found in rTHAs and rarely in primary surgeries. A weighty amount of data on the use of tantalum acetabular cups exist to prove their efficacy and superiority in these difficult cases. Starting in 2015, Meneghini et al. [28], in a multicentre clinical study, addressed patients with severe acetabular posterior column deficiency using tantalum buttress augments. None of the cases featured clinical or radiographic loosening nor needed reoperation, indicating that tantalum is a good substitute for the use of structural allografts or cages. Two years later, Diesel et al. [33] combined tantalum augments with lyophilized bovine xenograft, and demonstrated a 93.3% success rate concerning hip reconstruction in young patients with partial loss of the acetabular roof. In 2022, Alqwbani [56] et al. suggested satisfactory mid-term clinical and radiological outcomes in reconstructing major acetabular defects with the employment of tantalum shells and augments, whilst at the same time, Melnic et al. [57] reported encouraging short-term results with the combination of porous tantalum acetabular cup and a cemented monoblock dual-mobility acetabular component. Additionally, bispherical augments can be utilized as an alternative method in severe acetabular deficiency reconstruction with comparable clinical and radiological results to tantalum augments [59].

Several papers have compared tantalum with titanium cups on both the biomechanical and clinical levels. Theoretically, porous tantalum is characterized by an enhanced stability in higher loads [51] and is considered to provide substantial improvement in scores like the Harris Hip Score, WOMAC, and UCLA [27]. Concerning the survival [35] or the proximal migration [27] of the implant, there was no difference in primary or revision THAs. Different published studies connoted that there is small-scale avail in the utilization of porous tantalum over titanium in short- and mid-term follow-ups, except for lower risk of contamination, however, long-term results have favoured the tantalum cups so far [54]. Furthermore, a variety of articles have investigated the difference between tantalum and other porous implants, with the short-term results being the same regarding the survival and infection rates [37], but tantalum exhibited higher osseointegration [41]. Laaksonen et al. [37], measured the risk of re-revision between porous tantalum cups and other cementless materials, suggesting that there are no discrepancies in cup survival, but longer follow-ups are essential.

The majority of studies have reported excellent short- and mid-term outcomes with the utilization of porous tantalum components, like Siegmeth et al. [14] in 2009 and later on by Kamath et al. [21], with ameliorated clinical scores [24,46], but they all suggested that longer follow-up studies were required for more concrete inferences. In terms of long-term studies, concerning primary THAs, De Martino et al. [31], in a 15-year cohort study, presented improvement in HHS, with no radiographic evidence of loosening, while Konan et al. [3] also reported excellent functional results in uncemented revision THAs. Macheras et al. [34] in an 18-year follow-up study of 128 patients treated with monoblock tantalum acetabular cup, demonstrated dramatical amelioration in clinical scores, splendid radiographic outcomes with no evidence of migration, polyethylene wear or radiolucencies. Regarding the utilization of tantalum implants in revision THAs, the risk of failure is proportional to the acetabular defect [38]. In 2018, Lachiewicz et al. [39] indicated a durable fixation in mid-term follow-ups for complex acetabular defects, but the major problem appeared to be the percentage of hip joint dislocation, which was minimized with the use of larger femoral heads. At the same time, the authors proposed the employment of supplementary screw fixation of the shell [44] for increased stabilization of the cup in acetabular defects which also featured a small rate of complications like aseptic loosening and 2% infection rate with excellent long-term results. Later on, in 2022, Concina et al. [55] suggested that tantalum acetabular shells provide significant improvement in HHS and in restoring the hip centre of rotation in 25 patients with acetabular revisions and massive bone loss, reporting corresponding results with the study by Hsu et al. [58]. Moreover, the use of oblong cup with tantalum shells [47], and most importantly the combination of tantalum augments with antiprotrusion cages, was found to be a reliable technique for the restoration of the anatomic hip centre, preventing the superior migration of the implant and minimizing complication rates in massive acetabular defects cases [50]. Another consequential aspect is that tantalum cups, with or without additional wedges, can substantially reinstate the anatomical hip rotation centre and statistically significantly reduce the mean abduction angle of the acetabular cup [43].

Concerning failure figures, the literature data reccomend the employment of porous tantalum acetabular implants in both primary and revision THAs, as they present lower infection incidence [25,26] and radiolucencies [12,15,30]. This can be predominantly ascribed to its distinctive properties, such as a high porosity and coefficient of friction, leading to better osseointegration and elasticity to bone, while also reducing dead spaces. Although Theil et al. [46] warned that patients featuring tantalum acetabular cups should be monitored for aseptic loosening [11,14,21] that might require supplementary revision surgery, the vast majority of articles reported no signs of loosening radiographically [4,18,23,31]. Nevertheless, one of the major complications is indubitably THA dislocation. A noteworthy amount of research has been conducted on minimizing this risk. In 2014, Jain et al. [22] through a systematic review suggested that lower complication rates can be acquired with jumbo cups and tantalum systems. Later on, in 2018, Bruggeman et al. [42] indicated that dual-mobility cups cemented into porous tantalum shells in revision THA surgery present lower risks of dislocation, without reducing cup survival rates nor releasing more tantalum into blood stream. In addition, larger femoral heads can aid in enhancing the stability of challenging THAs [39]. Concerning the safety of the implant, Brüggemann et al. [49] stated that stable tantalum acetabular cups in THA feature limited blood concentrations of tantalum and no signs of T-cell activation.

A plethora of review articles can be found regarding the employment of tantalum acetabular cups in revision THAs. Starting in 2013, Issack et al. [20] denoted the excellent osseointegration of the implant in major bone loss surgeries, which might prove to be beneficial in the long-term if the cup growth occurs. Furthermore, the survival of augmentation cups appears to be increased in contrast to the traditional cage allografts. In the next year, Jain et al. [22], after reviewing 50 studies with a total of 2415 patients, propounded the use of jumbo or porous tantalum acetabular cups, as they exhibited lower complication rates, with the most common being aseptic loosening, THA dislocation, and infection. Later on, in 2019, Migaud et al. [45], through examining 28 articles, highlighted metallic reconstruction as the revolution for severe acetabular defects; however, further studies were needed with more than 10 years of follow-up for solid conclusions to be drawn. Last but not least, in the same year, Volpin et al. [47] summarized 50 articles with almost 3000 patients, suggesting that dislocation is the most common complication in revision THA surgeries, and out of the majority of implants, oblong cups and porous tantalum shells demonstrated the best survivorship rates. Contrariwise, only one review article by Rambani et al. [54] in 2022 is available concerning primary THA and it was a contrast between tantalum and titanium acetabular cups. More specifically, the first featured little advantage in short- to medium-term follow-up but exhibited promising long-term outcomes in cases with high-risk failure owing to mechanical loosening or infection. In a review of porous tantalum’s composition and its clinical application, Huang et al. [53] indicated that personalized tantalum-based implants have proven their clinical value and the results for modification methods that augment their bioactivity and antibacterial property are promising.

All in all, our meticulous review of the existing literature favours tantalum over titanium in terms of radiolucencies and osteolysis, infection rates, and osseointegration, as well as the survivorship of the tantalum implant, which reaches roughly 100% in over 10 years [20,52,55]. Finally, Hsu et al. [58] provided evidence of successful outcomes in patients with failed cage reconstruction, underlining the versatility of tantalum components in complex cases. In terms of clinical scores, no significant discrepancies were measured in short- and mid-term results, yet, regarding long-term results and cases with massive bone loss, the outcomes were propitious for tantalum.

## 5. Conclusions

Porous tantalum acetabular implants are broadly considered to provide promising results in complex THA cases with the possibility of infection and migration attributed to bone loss. In comparison with titanium, regarding radiolucencies, osteolysis, infection rates, osseointegration, and the survivorship of the implant, tantalum cups appear to prevail. Notwithstanding the fact that no statistically significant discrepancy has been observed concerning the clinical scores between them in straightforward cases, several published articles accentuate the predominance of tantalum cups in revision total hip arthroplasties and chiefly in complex revisions. Further pertinent research is requisite in order to corroborate the long-term employment of tantalum acetabular components in primary THA. Overall, whilst there have been instances of complications and there is a need for longer-term follow-ups, the vast majority of the studies propound positive outcomes associated with the utilization of porous tantalum acetabular implants in both primary and revision THAs. It appears to be a fruitful option in various clinical scenarios, with ongoing research aiming to optimize its use.

## Figures and Tables

**Figure 1 biomedicines-12-00959-f001:**
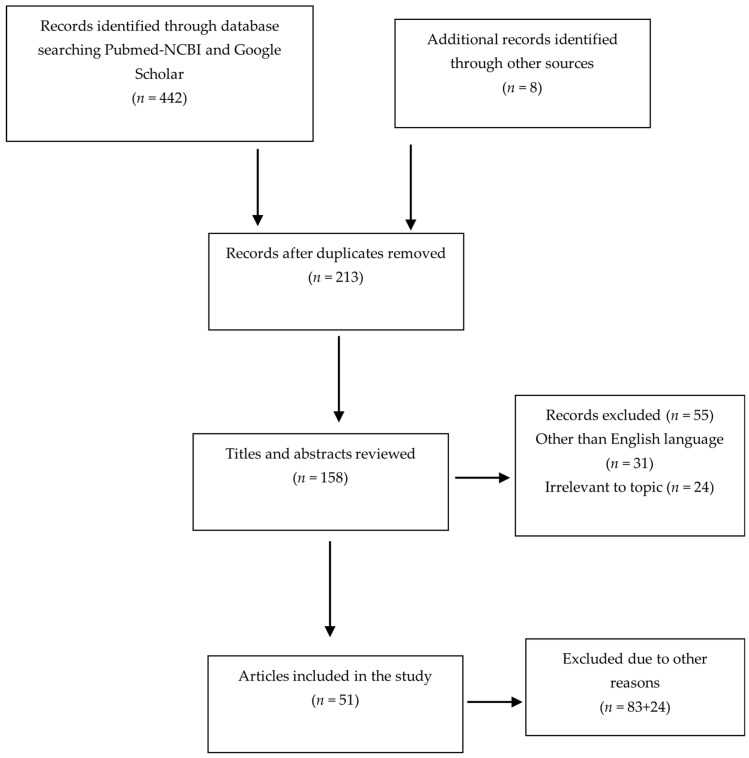
Flow chart of the systematic literature search according to PRISMA guidelines.

**Figure 2 biomedicines-12-00959-f002:**
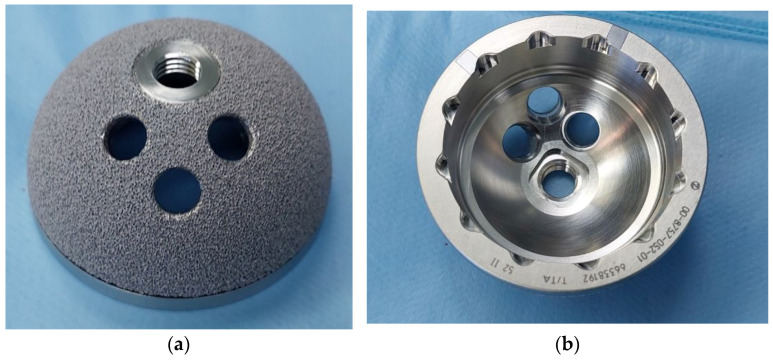
Pic 1: Porous tantalum acetabular cup. Image (**a**) is the outter porous tantalum part and (**b**) is the inner tantalum part of the acetabular cup (Courtesy of Mr Karampinas).

**Table 1 biomedicines-12-00959-t001:** Characteristics of the included studies.

Study (Year)/Reference	Title	Study Design	Results
Unger et al. (2005)/[11]	Evaluation of a porous tantalum uncemented acetabular cup in revision total hip arthroplasty: clinical and radiological results of 60 hips	Clinical study	Mean HHS significantly improved; Radiological excellent bone apposition and bone graft incorporation; Complications: seven dislocations and one revision for aseptic loosening Suitable for rTHA and warrants further study.
Gruen et al. (2005)/[12]	Radiographic evaluation of a Monoblock acetabular component: a multicentre study with 2- to 5-year results.	Multicentre cohort study	19% acetabular gaps which the majority filled in by the last follow-up; No evidence of radiolucencies; No evidence of lysis; No revisions for loosening; Encouraging short-term results; Longer follow-ups will be required.
Kim et al. (2008)/[13]	Porous tantalum uncemented acetabular shells in revision total hip replacement: two to four year clinical and radiographic results	Cohort study	Excellent early clinical and radiographic results with severe acetabular bone defects
Siegmeth et al. (2009)/[14]	Modular tantalum augments for acetabular defects in revision hip arthroplasty	Cohort study	Two revised for aseptic loosening; All quality-of-life parameters improved; Longer follow-up is required.
Macheras et al. (2009)/[15]	Eight- to ten-year clinical and radiographic outcome of a porous tantalum Monoblock acetabular component	Cohort study	Improvement in HHS, OHS; No radiographic evidence of gross polyethylene wear or, progressive radiolucencies, osteolytic lesions, acetabular fracture.
Li et al. (2009)/[16]	Radiographic appraisal between metal and bone interosculate backfill after total hip arthroplasty with trabecular metal cup	Cohort study	Improvement in HHS; 2% minor complications; No infection; No dislocation; No osteolysis or cup migration; Strong capacity of bone conductive and bone inducement.
Jafari et al. (2010)/[17]	Do tantalum and titanium cups show similar results in revision hip arthroplasty?	Cohort study	Tantalum is valuable in rTHA when a moderate-to-severe acetabular deficiency exists
Fernández-Fairen et al. (2010)/[18]	Revision of failed total hip arthroplasty acetabular cups to porous tantalum components: a 5-year follow-up study	Cohort study	Statistically significant improvement in mean HHS, WOMAC and Osteoarthritis Index scales; Radiographically stable; No re-revision for loosening.
Meneghini et al. (2010)/[19]	Mechanical stability of novel highly porous metal acetabular components in revision total hip arthroplasty	Biomechanical study	In vitro mechanical testing of tantalum metal show improved mechanical stability and osseointegration. Supplements the early successful clinical results particularly in the more complex and tenuous acetabular revisions
Flecher et al. (2010)/[4]	Do tantalum components provide adequate primary fixation in all acetabular revisions?	Clinical study	The mean Merle d’Aubigné score was 15.8; No radiolucent line; No revision for acetabular loosening; Three revisions for instability; Stable cementless fixation without compromising the centre of rotation; Longer follow-up is necessary.
Issack et al. (2013)/[20]	Use of porous tantalum for acetabular reconstruction in revision hip arthroplasty	Review Article	Excellent stability in acetabular revision; Stable reconstructions in major bone loss with good potential for bone ingrowth; Cup–cage reconstruction may be advantageous in the long term; Cup ingrowth increases the survivorship over traditional cage allograft constructs.
Kamath et al. (2013)/[21]	Total hip arthroplasty with porous metal cups following acetabular fracture	Cohort study	Improvement in WOMAC and UCLA score; One case of acetabular loosening; Longer follow-up is needed.
Jain et al. (2014)/[22]	Options for managing severe acetabular bone loss in revision hip arthroplasty. A systematic review	Systematic review	Jumbo cups and TM systems show lower complication rates; Most frequent complications: aseptic loosening, dislocation and infection.
Noiseux et al. (2014)/[23]	Uncemented porous tantalum acetabular components: early follow-up and failures in 613 primary total hip arthroplasties	Clinical study	4.4% revised; 1.2% acetabular cup removal; No aseptic loosening; High rates of initial stability and apparent ingrowth; Continued close follow-up is necessary to compare with second-generation porous-coated uncemented cups.
Moličnik et al. (2014)/[24]	Porous tantalum shells and augments for acetabular cup revisions	Clinical study	Improvement in HHS, WOMAC, and UCLA scores; No statistically significant differences in functional outcome; One revision due to traumatic dislocation; One radiographic lucent; No septic or aseptic failures; Sufficient primary stability in rTHA with acetabular bone loss; Longer-term follow-up studies are needed.
Tokarski et al. (2015)/[25]	Is tantalum protective against infection in revision total hip arthroplasty?	Comparative study	Lower incidence of failure and subsequent infection when used in patients with periprosthetic joint infection
Long et al. (2015)/[26]	Uncemented Porous Tantalum Acetabular Components: Early Follow-Up and Failures in 599 Revision Total Hip Arthroplasties	Cohort study	7.8% reoperation; 2.3% require cup removal due to infection; No aseptic loosening; Good initial stability and low re-operation rates.
Ayers et al. (2015)/[27]	Radiostereometric Analysis Study of Tantalum Compared with Titanium Acetabular Cups and Highly Cross-Linked Compared with Conventional Liners in Young Patients Undergoing Total Hip Replacement	Prospective Randomized Blinded study	Significant improvement in SF-36, WOMAC, UCLA, and HHS; No significant difference in proximal migration between the tantalum and titanium acetabular cups.
Meneghini et al. (2015)/[28]	Porous Tantalum Buttress Augments for Severe Acetabular Posterior Column Deficiency	Multicentre Clinical study	No cases of clinical or radiographic loosening; No reoperations; Tantalum seems a good substitute for the use of structural allografts or cages; Long term follow-up is required.
Nebergall et al. (2015)/[29]	Precision of radiostereometric analysis (RSA) of acetabular cup stability and polyethylene wear improved by adding tantalum beads to the liner.	Biomechanical study	Dispersion and number of beads are important in stability
Jeong et al. (2016)/[30]	Revision Total Hip Arthroplasty Using Tantalum Augment in Patients with Paprosky III or IV Acetabular Bone Defects: A Minimum 2-year Follow Up Study	Follow Up Study	Satisfactory clinical and radiographic outcomes in rTHA with severe acetabular bone defects of Paprosky type III or IV; 13.3% mild acetabular protrusion; 6.7% radiolucency around the acetabular cup without mechanical symptoms; One acute hematogenous infection.
Konan et al. (2016)/[3]	Porous tantalum uncemented acetabular components in revision total hip arthroplasty: a minimum ten-year clinical, radiological and quality of life outcome study	Cohort study	Excellent pain relief; Good functional outcomes; Patient satisfaction.
De Martino et al. (2016)/[31]	Long-Term Clinical and Radiographic Outcomes of Porous Tantalum Monoblock Acetabular Component in Primary Hip Arthroplasty: A Minimum of 15-Year Follow-Up	Cohort study	One cup revision for deep infection; No radiographic evidence of loosening, migration, or gross polyethylene wear; Improvement in HHS.
Lee et al. (2016)/[32]	Results of Total Hip Arthroplasty after Core Decompression with Tantalum Rod for Osteonecrosis of the Femoral Head	Cohort study	No significant differences in inclination or anteversion of acetabular cup; No evidence of osteolysis or subsidence of the femoral stem; Increased blood loss; One patient with squeaking.
Diesel et al. (2017)/[33]	Acetabular revision in total hip arthroplasty with tantalum augmentation and lyophilized bovine xenograft	Clinical study	Success rate for hip reconstruction in young patients with partial loss of the acetabular roof was 93.3%.
Macheras et al. (2017)/[34]	Survivorship of a Porous Tantalum Monoblock Acetabular Component in Primary Hip Arthroplasty with a Mean Follow-Up of 18 Years	Prospective study	HHS, OHS, and ROM dramatically improved; No evidence of migration, gross polyethylene wear, progressive radiolucencies, osteolytic lesions, or acetabular fractures; Excellent clinical and radiographic outcomes with no failures.
Vutescu et al. (2017)/[35]	Comparative survival analysis of porous tantalum and porous titanium acetabular components in total hip arthroplasty	Cohort study	Depending on the acetabular defect, there was no difference in survival between PoTi and PoTa acetabular components when used in primary or revision THA
Jenkins et al. (2017)/[36]	Minimum Five-Year Outcomes with Porous Tantalum Acetabular Cup and Augment Construct in Complex Revision Total Hip Arthroplasty.	Cohort study	3% aseptic loosening; 10% showed a radiolucent line between the trabecular metal shell and bone in DeLee and Charnley zone 3; 97% survivorship and maintained satisfactory hip function; In pelvic discontinuity consider of adding adjunctive fixation or alternative techniques.
Laaksonen et al. (2017)/[37]	Does the Risk of Rerevision Vary Between Porous Tantalum Cups and Other Cementless Designs After Revision Hip Arthroplasty?	Clinical study	No difference in cup survival in rTHA; No benefit in survival with rerevision for infection; Longer follow-up are needed.
Ling et al. (2018)/[38]	The Use of Porous Tantalum Augments for the Reconstruction of Acetabular Defect in Primary Total Hip Arthroplasty	Cohort Study	HHS, OHS, University of California Los Angeles activity scale, and Short Form-12 score improved significantly; Anatomical cup placement; No aseptic loosening, cup and augment migration, screw breakage, or presence of hip infection; Satisfactory radiographic outcomes in acetabular defect in primary THA.
Lachiewicz et al. (2018)/[39]	Tantalum Components in Difficult Acetabular Revisions Have Good Survival at 5 to 10 Years: Longer Term Follow-up of a Previous Report	Cohort study	15% dislocation and 10% required reoperation; Durable fixation at midterm follow-up in complex acetabular revisions; Propose to use larger femoral head to minimize dislocation; Further longer and multicentre studies are necessary.
Solomon et al. (2018)/[40]	The Stability of the Porous Tantalum Components Used in Revision THA to Treat Severe Acetabular Defects: A Radiostereometric Analysis Study	Biomechanical study	Acceptable early migration in rTHA with severe acetabular defects; Good long-term survivorship; Improvement of acetabular fixation with inferior screws.
Bondarenko et al. (2018)/[41]	Comparative analysis of osseointegration in various types of acetabular implant materials	Clinical study	Porous tantalum trabecular metal implants exhibit higher osseointegration
Brüggemann et al. (2018)/[42]	Do dual-mobility cups cemented into porous tantalum shells reduce the risk of dislocation after revision surgery?	Clinical study	Lower risk of dislocation without reducing the cup survival nor releasing more tantalum
Barros et al. (2019)/[43]	Recovery of the Hip Rotation Centre with Tantalum in Revision Arthroplasty	Clinical study	RTHA with tantalum cups, associated or not with addition wedges, significantly recovered the anatomical rotation centre of the hip; Statistically significant decrease in the mean abduction angle of the acetabular cup.
Löchel et al. (2019)/[44]	Reconstruction of acetabular defects with porous tantalum shells and augments in revision total hip arthroplasty at ten-year follow-up	Cohort study	5.6% had revision due to aseptic loosening; 1.9% had revision for infection; Mean HHS improved significantly; Excellent long-term results; Supplementary screw fixation of the shell is suggested; Alternative techniques in pelvic discontinuity.
Migaud et al. (2019)/[45]	Acetabular reconstruction using porous metallic material in complex revision total hip arthroplasty: A systematic review	Systematic review	Metallic reconstruction, is a progress in the treatment of severe bone defects in rTHA; Indications: failure of allograft associated with a cage in Paprosky type 3 defects, especially in pelvic discontinuity.
Theil et al. (2019)/[46]	A single centre study of 41 cases on the use of porous tantalum metal implants in acetabular revision surgery	Cohort study	Good to excellent short- and mid-term functional results; 19.5% of aseptic loosening; 4.9% infection; Higher rate of failure with major bone loss defects.
Volpin et al. (2019)/[47]	Reconstruction of failed acetabular component in the presence of severe acetabular bone loss: a systematic review	Systematic review,	Most common complication is dislocation rTHA depending on the type of bone loss Oblong cups and tantalum shells show the best survivorship,
Miettinen et al. (2020)/[48]	Revision hip arthroplasty using a porous tantalum acetabular component	Retrospective study	High porosity, high frictional characteristics, low migration rate and low modulus of elasticity; Risk factors for dislocation: Malposition and small head size (28 mm).
Brüggemann et al. (2020)/[49]	Safety of Use of Tantalum in Total Hip Arthroplasty	Clinical study	Stable tantalum cups have low blood concentrations of tantalum; No signs of T-cell activation typical of ALVAL.
Baecker et al. (2020)/[50]	Tantalum Augments Combined with Antiprotrusion Cages for Massive Acetabular Defects in Revision Arthroplasty	Retrospectively clinical study	All clinical outcome scores significantly improved postoperatively; Complications (10%):2 re-revisions for aseptic aetiologies and 1 for loosening; Reliable technique to restore the anatomic hip centre and prevent superior migration and provides a bony ingrowth surface; Long-term follow-up is required before the technique is widely adapted.
Beckmann et al. (2020)/[51]	Comparison of the Primary Stability of Porous Tantalum and Titanium Acetabular Revision Constructs	Biomechanical study	Tantalum may provide a greater primary stability at higher loads than titanium; Further clinical studies are necessary.
Cassar-Gheiti et al. (2021)/[52]	Midterm Outcomes After Reconstruction of Superolateral Acetabular Defects Using Flying Buttress Porous Tantalum Augments During Revision Total Hip Arthroplasty	Cohort Study	Excellent implant survivorship and favourable clinical outcomes
Huang et al. (2021)/[53]	The Clinical Application of Porous Tantalum and Its New Development for Bone Tissue Engineering	Review article	Personalized porous Ta-based implants have shown their clinical value; Promising results with modification methods that enhance their bioactivity and antibacterial property.
Rambani et al. (2022)/[54]	Tantalum Versus Titanium Acetabular Cups in Primary Total Hip Arthroplasty: Current Concept and a Review of the Current Literature	Review article	High risk of failure or mechanical loosening; Has lower risk of failure and contamination; Promising long-term results; Little advantage in short- to medium-term follow-up.
Concina et al. (2021)/[55]	Can porous tantalum acetabular cups and augments restore the hip centre of rotation in revision hip arthroplasty? Long-term results	Clinical study	Statistically significant improvement in HHS and Hip ROM; Kaplan–Meier survivorship of 100%; Complications: three dislocations, four asymptomatic heterotopic ossifications, and one partial reabsorption of greater trochanter; Valid solution in acetabular revisions for addressing massive bone defects and restoring the hip centre of rotation.
Alqwbani et al. (2022)/[56]	Porous tantalum shell and augment for acetabular defect reconstruction in revision total hip arthroplasty: a mid-term follow-up study	Follow-up study	WOMAC pain score 90.5 and WOMAC function 88.3; Mean OHS 89.2; Satisfactory mid-term clinical and radiological outcomes in reconstructing major acetabular defects without Paprosky IIIA defect in rTHA.
Melnic et al. (2022)/[57]	Treatment of Severe Acetabular Bone Loss Using a Tantalum Acetabular Shell and a Cemented Monoblock Dual Mobility Acetabular Cup	Clinical study	Encouraging short-term results
Hsu et al. (2022)/[58]	Acetabular Revision Surgery with Tantalum Trabecular Metal Acetabular Cup for Failed Acetabular Cage Reconstruction with Bone Allografts: A Retrospective Study with Mid- to Long-Term Follow-Up	Retrospective Study	Good results in patients with failed cage reconstruction with bone allografts
Li et al. (2022)/[59]	Modular revision strategy with bispherical augments in severe acetabular deficiency reconstruction	Clinical study	Bispherical augments show comparable clinical and radiological results with tantalum augments; Can be used as alternative method in severe acetabular deficiency reconstruction.

## Data Availability

Publicly available datasets were analyzed in this study. This data can be found here in https://pubmed.ncbi.nlm.nih.gov/.

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
