# Peer review of "Porous Tantalum Acetabular Cups in Primary and Revision Total Hip Arthroplasty: What Has Been the Experience So Far?—A Systematic Literature Review"

_biomedicines, 2024, doi:10.3390/biomedicines12050959_

Round 1
Reviewer 1 Report
Comments and Suggestions for Authors
The manuscript # biomedicines-2929468 entitled "Porous tantalum acetabular cups in primary and revision total hip arthroplasty: what has been the experience so far? – A systematic literature review" has been submitted for publication in the journal Biomedicines (MDPI).
This article is a comprehensive review of 61 articles of the scientific literature dealing with the use of porous tantalum as highly valuable material for total hip arthroplasty applications.
The authors thoroughly introduce the topic to describe the history of the clinical use of tantalum as bone implant. Results dealing with radiological improvements, mechanical stability, osseointegration, and biomechanical studies are described and discussed. The authors compare the properties of tantalum to those of titanium, the gold standard of bone implants. Better mechanical and biological properties of tantalum are particularly highlighted.
In my opinion, this review of the literature is really interesting. The study is well conducted, the presentation is clear, including a valuable table compiling the studied references. The whole manuscript gathers a lot of accurate information on a modern research topic with relevant clinical perspectives.
I recommend to this review article to be accepted for publication in the journal Biomedicines.
Author Response
Thank you very much for your opinion on our article. We appreciate your kind and truthful review.
Reviewer 2 Report
Comments and Suggestions for Authors
- The Abstract lacks a section of Results-
- “Keyword search terms were: tantalum AND titanium AND hip arthroplasty” But only 4 articles deal with tantalum and titanium comparison. Is this right?
- One of the inclusion criteria is:
“Direct comparison between tantalum acetabular cups and conventional titanium acetabular cups employed in total hip arthroplasty” But only 4 articles deal with tantalum and titanium comparison. Is this right?
- The Flow chart (fig. 1) is not coherent with the description written in the same page. Please improve this.
- In some sections, authors did not indicate in the text all the mentioned references. As an example, in section 3.1. Clinical and Radiological Results – authors refer to as Sixteen studies but only some of the references appear referred in the text (11 references). Please check this in all sections.
Author Response
Thank you for your review. The articles comparing just titanium and tantalum are 4 but there are 16 more comparing those two metals with others. So 20 in total comparing different prosthesis but our goal was comparing only titanium and tantalum. To end with, there is a whole section of Results with 6 subcategories. We will improve the flow chart of the article and elucidade the references appearance throughout the manuscript.
Reviewer 3 Report
Comments and Suggestions for Authors
The article is well-written, has correct English language, explanatory figures and tables, clear analysis of the state of the art, and results that are consistent with the findings presented. It provides valuable insights into the use of porous tantalum acetabular cups in hip arthroplasty, highlighting their clinical improvement, stability, and long-term outcomes. The systematic review methodology ensures a comprehensive evaluation of published studies up to date (articles are sufficiently recent). The results suggest that porous tantalum is a promising option in revision total hip arthroplasty, providing clinical improvement and radiological stability. Ongoing research and longer follow-up periods are needed to further validate and refine its use.
However, it should it be noted that it needs correction in the citation format in line with Biomedicines guidelines. For instance, in line 39, instead of using the notation (1),(2), it is recommended to use [1,2] to comply with the citation standards in research.
Author Response
Thank you for your response and for your nice and honest opinion. We will correct the citation format in the article
Round 2
Reviewer 2 Report
Comments and Suggestions for Authors
The authors addressed my comments.